

# Transitions in Xenes between excitonic, topological and trivial insulator phases: Influence of screening, band dispersion and external electric field

Olivia Pulci[1], Paola Gori[2], Davide Grassano[3],
Marco D'Alessandro[4] and Friedhelm Bechstedt[5]

**1** Department of Physics, and INFN, University of Rome Tor Vergata,
Via della Ricerca Scientifica 1, I-00133 Rome, Italy
**2** Department of Industrial, Electronic and Mechanical Engineering, Roma Tre University,
Via della Vasca Navale 79, I-00146 Rome, Italy
**3** Theory and Simulation of Materials (THEOS), École Polytechnique Federale de Lausanne,
1015 Lausanne, Switzerland
**4** Istituto di Struttura della Materia-CNR (ISM-CNR), Division of Ultrafast Processes
in Materials (FLASHit), Via del Fosso del Cavaliere 100, 00133 Rome, Italy
**5** Institut für Festkörpertheorie und -optik, Friedrich-Schiller-Universität Jena,
Max-Wien-Platz 1, 07743 Jena, Germany

## Abstract

Using a variational approach, the binding energies $E_b$ of the lowest bound excitons in Xenes under varying electric field are investigated. The internal exciton motion is described both by Dirac electron dispersion and in effective-mass approximation, while the screened electron-hole attraction is modeled by a Rytova-Keldysh potential with a 2D electronic polarizability $\alpha_{2D}$. The most important parameters as spin-orbit-induced gap $E_g$, Fermi velocity $v_F$ and $\alpha_{2D}$ are taken from *ab initio* density functional theory calculations. In addition, $\alpha_{2D}$ is approximated in two different ways. The relation of $E_b$ and $E_g$ is ruled by the screening. The existence of an excitonic insulator phase with $E_b > E_g$ sensitively depends on the chosen $\alpha_{2D}$. The values of $E_g$ and $\alpha_{2D}$ are strongly modified by a vertical external electric bias $U$, which defines a transition from the topological into a trivial insulator at $U = E_g/2$, with the exception of plumbene. Within the Dirac approximation, but also within the effective mass description of the kinetic energy, the treatment of screening dominates the appearance or non-appearance of an excitonic insulator phase. Gating does not change the results: the prediction done at zero electric field is confirmed when a vertical electric field is applied. Finally, Many-Body perturbation theory approaches based on the Green's function method, applied to stanene, confirm the absence of an excitonic insulator phase, thus validating our results obtained by *ab initio* modeling of $\alpha_{2D}$.

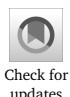

## 1 Introduction

Excitonic insulators (EIs) arise from the spontaneous formation of bound electron-hole pairs, the excitons, in semiconductors with small fundamental gap $E_g$ but large binding energy $E_b$ of the lowest-energy excitonic excitation [1–4]. Formally, their appearance can be characterized by the relation $E_b > E_g$. Then, the electronic system is unstable against the formation of charge or spin density waves. A close formal similarity between the EI phase and the superconducting state has been predicted. However, physical properties of the two states of matter are different, e.g. no Meissner effect should be observable in an EI. In the three-dimensional (3D) case, experimental evidence has been found for III-V semiconductor quantum well systems, e.g. InAs/GaSb heterostructures [5], or layered semiconductors such as $Ta_2NiSe_5$ [6] and 1T-TiSe$_2$ [7].

Among two-dimensional (2D) systems the existence of the EI phase should be more likely, because of the reduced screening in two dimensions and the consequent large exciton binding energy $E_b$ [8,9]. Indeed, a theoretical prediction of the EI has been recently made for monolayer transition-metal dichalcogenides (TMDCs) such as $1T'-MoS_2$ [10]. Interestingly, these materials may represent a topological insulator (TI) [11] and enable the realization of the quantum spin Hall (QSH) effect at room temperature [12,13]. An outstanding candidate for the observation of the EI and TI phases with "topological excitons" is the $1T'-WTe_2$ monolayer system [14]. The band inversion and the spin-orbit coupling (SOC) are responsible for its non-trivial topology. By means of transport measurements, indications for the TI phase have been found [15,16]. Other TMDCs in $1T'$ structure such as $MoS_2$ and $WS_2$ are also TIs and, because of the small band gap, are candidates for the EI phase [17]. Double layers of TMDC, e.g. the combination $WSe_2/MoSe_2$, seem to be also candidates for strongly correlated EI phases [18]. Recently, heterojunction Moiré superlattices made of $WS_2/WSe_2$ have been identified as EIs [19,20].

The graphene-like but buckled 2D allotropes of group-IV elements, the Xenes silicene, germanene, stanene and plumbene, with a small SOC-induced fundamental gap should be also outstanding candidates for the observation of the EI phase [21]. Because of the band inversion, these honeycomb materials are also TIs [21–26] with a static QSH conductivity nearly equal to the conductance quantum [17, 24]. The existence of an EI phase in silicene, germanene and stanene have been first studied by Brunetti et al. [27, 28] in the framework of the effective-mass approximation (EMA) [29] of the conduction and valence bands around $K$ and $K'$ and the screened Rytova-Keldysh potential [30, 31] of the electron-hole attraction with a bulk-like screening. They also predicted a phase transition in freestanding monolayer Xenes from the EI phase to the trivial semiconducting phase driven by an external vertical electric field and including the effect of embedding dielectric materials. Corresponding phase transitions between the TI phase and the trivial one under the action of external bias voltages have been also theoretically predicted for Xenes [17, 24, 32].

In this paper we investigate the existence of the EI phase in Xenes in a more complete manner. In the analytic description of the formation of excitons at the absorption edge, we fully account for the dispersion relation of the massive Dirac fermions, electrons and holes, at the lowest conduction bands and highest valence bands near the Dirac points $K$ or $K'$. The statically screened Coulomb attraction of Dirac electrons and holes is described as a Rytova-Keldysh potential. However, in the original Rytova-Keldish model the screening is introduced through a bulk dielectric constant. Here, we model the screening by the static electronic polarizability $\alpha_{2D}$ of the 2D system. The parameter $\alpha_{2D}$ is taken from *ab initio* calculations of the optical conductivities in the limit of vanishing frequency. Moreover, also an analytical model for the $\alpha_{2D}$ is applied, derived from a four-band tight-binding model. We show that the different screenings, bulk or 2D-derived, rule the existence or not of the excitonic insulator phase.

Further, the influence of the more accurate band dispersion is studied by comparison with results from EMA. The Xenes are also studied under the action of an external electric gate field. The corresponding phase transformation between the TI phase, for low field strength, and the trivial phase, above a critical bias, is compared with the occurrence of a EI phase. Finally, for stanene, we compare the Rytova-Keldysh exciton binding energies with the *ab initio* value derived from the solution of the Bethe-Salpeter equation.

## 2 Theoretical and computational methods

### 2.1 Atomic geometry, electronic and optical properties

The basic atomic geometries and electronic structures are obtained in the framework of the density functional theory (DFT) [33, 34], as implemented in the QUANTUM ESPRESSO package [35, 36], with the semilocal Perdew-Burke-Ernzerhof (PBE) exchange-correlation (XC) functional [37], and a plane-wave expansion of the single-particle wave functions up to energies of 90 Ry. A 3D superlattice arrangement is applied to simulate the isolated Xenes sheets. Correspondingly, in the ground-state calculations, the Brillouin Zone (BZ) sampling is performed by a 12×12×1 **k**-point Monkhorst-Pack [38] mesh centered at Γ. The calculations of optical and dielectric properties have been performed with more dense 600×600×1 (300×300×1 for plumbene) **k**-point meshes. For silicene, given the extremely small gap of 1.5 meV, we find that even denser meshes are required to obtain properly converged results. In order to overcome hardware and code limitations, we make use of progressively denser grids cropped around the $K$ point in order to determine the optical properties in a low energy range. We find that the results up to 12 meV are converged with a 12000×12000×1 grid and a crop

Table 1: Structural (lattice constant $a$, buckling $\Delta$), electronic (direct band gap $E_g$, Fermi velocity $v_F$, effective interband mass $\mu$), and dielectric (static 2D polarizability $\alpha_{2D}$) parameters of slightly buckled Xenes derived in DFT-PBE. In addition to the resulting $\alpha_{2D}$(DFT) values (see Eq. (11)), two other values, $\alpha_{2D}$(bulk)$=\delta\epsilon/4\pi$ from a bulk-like approach and $\alpha_{2D}$(model) (see Eq. (12)) are also listed.

| parameter | silicene | germanene | stanene | plumbene |
|---|---|---|---|---|
| $a$ (Å) | 3.874 | 4.045 | 4.673 | 4.958 |
| $\Delta$ (Å) | 0.44 | 0.68 | 0.86 | 0.98 |
| $E_g$ (meV) | 1.5 | 24.2 | 77.2 | 491 |
| $v_F$ ($10^6$m/s) | 0.53 | 0.52 | 0.47 | 0.45 |
| $\mu$ (m$_e$) | 0.00025 | 0.00394 | 0.01537 | 0.10664 |
| $\alpha_{2D}$(bulk) (Å) | 3.8 | 5.7 | 9.5 | $\infty$ |
| $\alpha_{2D}$(model) (Å) | 1909.1 | 126.2 | 39.6 | 6.2 |
| $\alpha_{2D}$(DFT) (Å) | 2500 | 149.1 | 44.9 | 7.7 |

radius of 0.01 Å$^{-1}$, while a 6000×6000×1 and 2400×2400×1 grids with crop radii of 0.02 and 0.06 Å$^{-1}$ are used to obtain converged properties up to 200 and 450 meV respectively. Above this threshold, the optical properties are already converged with the 600×600×1 grid.

Results obtained for the low buckled freestanding Xenes are summarized in Table 1 which reports the 2D lattice constant $a$, the buckling parameter $\Delta$, the SOC-induced (direct) fundamental gap $E_g$ at $K$ and $K'$, the Fermi velocity $v_F$ of the Dirac bands in the vicinity of $K$ and $K'$, and their interband mass $\mu = E_g/(2v_F)^2$. While silicene, germanene, and stanene are direct semiconductors, plumbene exhibits an indirect gap of 0.42 eV (from almost $\Gamma$ to $K/K'$), slightly smaller than the direct $K/K'$ gap $E_g = 0.49$ eV. The results are compatible with other values derived within the DFT-PBE framework for silicene, germanene, and stanene [21, 29, 39, 40] and for plumbene [41].

We have performed calculations of the $\mathbb{Z}_2$ topological invariant using the Z2 pack software [42], where the evolution of hybrid Wannier centers is implemented also for non-centrosymmetric materials [43]. We found that $\mathbb{Z}_2 = 0$, i.e. plumbene is a trivial insulator, in agreement with other plumbene studies [26, 44, 45]. This is in contrast to the other Xenes silicene, germanene and stanene, which are topological insulators for fields below the critical strength.

It is well known that Kohn-Sham band structures [34] systematically underestimate gap energies and interband distances [46]. The account of quasiparticle effects not only opens gaps but also modifies the band dispersion by increasing the Fermi velocity $v_F$ of the Dirac bands. Indeed, in a simplified quasiparticle approach, e.g. using the hybrid XC functional of Heyd, Scuseria and Ernzerhof HSE06 [47, 48], an increase of the SOC-induced gaps and of the Fermi velocities occurs [29]. The explicit inclusion of quasiparticle effects on the excitons and on the electronic bands is extremely demanding for small-gap 2D systems, because of the slow convergence of the optical properties with the number of k-points. Anyway, quasiparticle and excitonic effects tend to cancel each other [49, 50].

## 2.2 Two-particle excitations: Excitons

Electron-hole pair excitations with electrons in a conduction band $\varepsilon_c(\mathbf{k})$ and holes in a valence band $\varepsilon_v(\mathbf{k})$ can be described by Bethe-Salpeter equation (BSE) with attractive statically screened Coulomb interaction $\hat{W}$ and a bare repulsive electron-hole exchange [46]. Such a description is usually employed to compute excitonic states from first principles with Bloch bands and Bloch states taken from an approximate quasiparticle description [8, 9, 51, 52]. Such an

approach has been used to calculate the 2D optical conductivity in a wide energy range also for Xenes, e.g. for freestanding silicene [49, 53]. However, because of the small fundamental gap and the pronounced linear bands near $K$ and $K'$ points, an extremely dense **k**-point sampling around the BZ corner points is required to correctly describe the lowest-energy pair excitations and the possible occurrence of the EI phase. Modeling the electronic and optical properties of Xenes can be the way to overcome this issue.

### 2.2.1 Modeling: Single-particle bands

Because of the SOC-induced gap $E_g$ the bands at $K$ or $K'$ are parabolic just in a narrow region around the Dirac points but the effective mass symmetry between electron and hole is preserved. In the presence of an external vertical gate electric field $F$, characterized by a potential energy difference $U = eF\Delta/2$, the band energies are [17, 21, 23, 29]

$$\varepsilon_{\xi vs}(\boldsymbol{\kappa}) = v\left[\left(U - \xi s\frac{1}{2}E_g\right)^2 + \hbar^2 v_F^2 \kappa^2\right]^{\frac{1}{2}}, \tag{1}$$

with the valley index $\xi = +, -$, the conduction or valence band $v = +, -$, the spin orientation $s = +, -$, and the wavevector variation $\boldsymbol{\kappa} = \mathbf{k} - \mathbf{k}_{K/K'}$. The corresponding field-modified gap is

$$E_g(U) = \left|E_g - 2|U|\right|, \tag{2}$$

while the bands exhibit field-induced splittings $\Delta\varepsilon = 2|U|\theta(E_g - 2|U|) + E_g\theta(2|U| - E_g)$. At the critical field strength, $U_{\mathrm{crit}} = \frac{1}{2}E_g$, which defines the transition between the TI and trivial phase of the Xenes [17, 23, 24, 32], with the exception of plumbene where this transition does not occur.

### 2.2.2 Modeling: Two-particle Hamiltonian

In a two-band model, the 2D excitonic Hamiltonian in real space can be approximated as [54]:

$$\left\{E_{cv}(-i\boldsymbol{\nabla}_{\mathbf{x}}) + \hat{W}(\mathbf{x})\right\}\Phi_0(\mathbf{x}) = E_0\Phi_0(\mathbf{x}), \tag{3}$$

where $\hat{W}(\mathbf{x})$ is the the statically screened Coulomb potential, $\mathbf{x}$ the in-plane electron-hole distance, and $E_0 = E_g - E_b$ the lowest electron-hole excitation energy, with $E_b$ the exciton binding energy. $E_{cv}(\boldsymbol{\kappa}) = \varepsilon_c(\boldsymbol{\kappa}) - \varepsilon_v(\boldsymbol{\kappa})$ denotes the interband energy defined as difference between the lowest conduction band $\varepsilon_c(\boldsymbol{\kappa})$ and the highest valence band $\varepsilon_v(\boldsymbol{\kappa})$ around the $K$ or $K'$ point.

Going from reciprocal space to real space, the wavevector $\boldsymbol{\kappa}$ is formally replaced by the operator $-i\boldsymbol{\nabla}_{\mathbf{x}}$. In the limit of Wannier-Mott excitons [46, 54] the electron-hole exchange interaction is negligible. Therefore, in Eq. (3) only the screened electron-hole attraction $\hat{W}$ appears.

For Dirac systems, studied here, the linearity of the bands and spin degeneracy in Eq. (1) give, in absence of an external field,

$$E_{cv}(\boldsymbol{\kappa}) = \sqrt{E_g^2 + (2\hbar v_F \kappa)^2}, \tag{4}$$

which, for small wavevectors $|\boldsymbol{\kappa}|$, delivers the so-called effective mass approximation (EMA):

$$E_{cv}(\boldsymbol{\kappa}) = E_g + \frac{\hbar^2}{2\mu}\kappa^2, \tag{5}$$

with $\mu = E_g/(2v_F)^2$ the exciton reduced mass and $v_F$ the Fermi velocity. The differential operator in Eq. (3), defined by a power series, can be split into

$$E_{cv}(-i\boldsymbol{\nabla}_{\mathbf{x}}) = E_g + \hat{T}(\mathbf{x}), \tag{6}$$

with the generalized kinetic energy operator for Dirac systems

$$\hat{T}(\mathbf{x}) = \sqrt{E_g^2 - (2\hbar v_F \boldsymbol{\nabla}_\mathbf{x})^2} - E_g \,. \tag{7}$$

In an extremely narrow region around $K$ or $K'$, Eq. (7) reduces to the EMA expression

$$\hat{T}(\mathbf{x}) \approx -\frac{\hbar^2}{2\mu} \boldsymbol{\nabla}_\mathbf{x}^2 \,. \tag{8}$$

Most important for the description of the lowest exciton bound state in Xenes, i.e., atomic layers, is the screened Coulomb attraction $\hat{W}(\mathbf{x})$ between electrons and holes. In the true 2D limit of sheets embedded in a dielectric with dielectric constant $\bar{\epsilon}$, this screened potential in Fourier space is given by [8, 55, 56]

$$W(\boldsymbol{\kappa}) = -\frac{2\pi e^2}{\bar{\epsilon}|\boldsymbol{\kappa}|} \frac{1}{1 + 2\pi\alpha_{2D}|\boldsymbol{\kappa}|/\bar{\epsilon}} \,, \tag{9}$$

with $\alpha_{2D}$ as the electronic polarizability of a true 2D electron gas. The screening can be also described within a quasi-3D approach, e.g. a quantum well structure. In this limit, electrons and holes are excited in a semiconductor of thickness $\delta$ and dielectric constant $\epsilon$ that is embedded by infinitely thick barrier layers with dielectric constant $\bar{\epsilon}$. In Fourier space, in the limit $\delta \to 0$, formally the same wavevector dependence as in Eq. (9) is obtained [31, 32, 46, 56]. However, instead of the 2D polarizability $\alpha_{2D}$, the product $\epsilon\delta/4\pi$ appears. Therefore, one may identify $\alpha_{2D}(\text{bulk}) = \epsilon\delta/4\pi$ and call this as bulk-like model.

In general, the screened potential in 2D of Eq. (9) is approximated assuming a constant electronic polarizability $\alpha_{2D}$ of the sheet and an averaged dielectric constant $\bar{\epsilon}$ of the embedment. In real space Eq. (9) transforms into the Rytova-Keldysh form [30, 31]

$$\hat{W}(\mathbf{x}) = -\frac{\pi}{2} \frac{e^2}{\rho_0 \bar{\epsilon}} \left[ H_0\left(\frac{\rho}{\rho_0}\right) - N_0\left(\frac{\rho}{\rho_0}\right) \right], \tag{10}$$

with the planar distance $\rho = |\mathbf{x}|$, the characteristic screening radius $\rho_0 = 2\pi\alpha_{2D}/\bar{\epsilon}$, and the Bessel functions of second kind, the Struve function $H_0$ and the Neumann function $N_0$. In the cases where the Xenes are free standing, $\bar{\epsilon} = 1$ holds.

In the limit of small thicknesses $\delta$ of the 2D object, in both cases of description of the screening, (i) starting from the 2D character of the electronic system or (ii) starting from a bulk semiconductor with a defined dimensionless bulk dielectric constant $\epsilon$, the expression of the screened potential $\hat{W}$ (Eq. (10)) remains the same, despite the completely different character of the screening.

### 2.2.3 Static electronic polarizability $\alpha_{2D}$

Within the bulk-like screening approximation (ii), applying the $\epsilon$ and $\delta$ values used by Brunetti et al. [28], one finds extremely small polarizability values $\alpha_{2D}(\text{bulk}) = 3.8, 5.7, 9.5$ Å for silicene, germanene, and stanene (see Table 1). Here, because of the metallic character of lead, an infinite dielectric constant is chosen in the plumbene case. It formally leads to $\alpha_{2D}(\text{bulk}) = \infty$. We study this screening approximation for the purpose of comparison.

In the other class of screening (i), starting directly from the 2D character of the electronic system, the electronic polarizability is directly calculated using band energies and 2D Bloch functions of the Xenes. We do so using two methods to determine $\alpha_{2D}$. The static electronic polarizabilities $\alpha_{2D}$ of 2D sheets can be generated within *ab initio* DFT calculations in independent (quasi)-particle approximation [46], sometimes also called, in a less accurate manner, random phase approximation (RPA) [54].

We call the results $\alpha_{2D}$(DFT). Dielectric and screening properties can be described by the in-plane optical conductivity $\sigma(\omega)$ in the low-frequency limit (see Supplemental Material and [21]). The static electronic polarizability $\alpha_{2D}$ is given by

$$\alpha_{2D} = -\lim_{\omega \to 0} \frac{1}{\omega} Im\sigma(\omega) = \lim_{\omega \to 0} \frac{L}{4\pi} (Re\epsilon_\parallel^{SL}(\omega) - 1), \tag{11}$$

with the in-plane component of the dielectric function $\epsilon_\parallel^{SL}(\omega)$ of a superlattice (SL) of 2D sheets in a distance $L$, that is used in numerical calculations. The explicit *ab initio* calculations of $\alpha_{2D}$(DFT) require an extremely dense **k**-point mesh. Because of the high **k**-point density used, i.e., more oscillators, the values $\alpha_{2D}$(DFT) in Table 1 are significantly larger than the values given in Ref. [21].

The in-plane optical conductivity $\sigma(\omega)$ can be also calculated applying the model band structure Eq. (1), only valid around the $K$ and $K'$ points. The tight-binding method resulting in the bands in Eq. (1) gives, together with Eq. (11), a clear analytical relation between $\alpha_{2D}$ and the fundamental gap $E_g$ as [21]

$$\alpha_{2D} = \frac{2}{3\pi} \frac{e^2}{E_g}. \tag{12}$$

We will call the resulting polarizabilities $\alpha_{2D}$(model). Expression (12) suggests that the influence of an electric field can be included using Eq. (2). The extraordinary advantage of the model expression in Eq. (12) for the static 2D electronic polarizability is the drastic reduction of the numerical efforts. Only the calculation of the fundamental gap of the 2D system is needed. As shown in Table 1, the two 2D approximations for $\alpha_{2D}$, $\alpha_{2D}$(DFT) and $\alpha_{2D}$(model), give similar values. This is a remarkable result, since the numerical calculation of $\alpha_{2D}$(DFT) can be very heavy for Dirac systems with very small gaps: thousands and thousands of **k**-points are needed to sample the Brillouin zone near the Dirac point to get well converged optical constants. On the contrary, the analytical $\alpha_{2D}$(model) depends only on the value of the electronic gap, hence on the energy bands at one single **k**-point. The values $\alpha_{2D}$(model) remain somewhat smaller than the DFT ones because only the lowest-energy interband oscillators are taken into account. Both follow a chemical trend with the inverse fundamental gap $1/E_g$ of the 2D system, while the bulk-like approximation according to Rytova and Keldysh [30, 31] for $\alpha_{2D}$(bulk) follows the inverse fundamental (or Penn) gap of the corresponding bulk elemental material. The resulting trends are therefore opposite along the series Si→Ge→Sn→Pb. This puzzling behavior is caused by the fact that the 2D gap arises from spin-orbit interaction, which increases with increasing atomic number, while the Penn gap, dominating the electronic polarization ($\epsilon - 1$) of the corresponding 3D system, is inverse to the square of the atomic distances.

### 2.2.4 Binding energy

In order to compute the binding energy $E_b$ of the lowest-energy exciton, we apply a variational method [56]. $\Phi_0(\mathbf{x})$ in Eq. (3) is replaced by a $1s$ trial wave function $\Phi_0(\mathbf{x}) \propto e^{-2\lambda\rho/a_{ex}}$, with the exciton radius $r_{ex} = a_{ex}/(2\lambda)$ defined by the variational parameter $\lambda$ and the Wannier-Mott exciton radius $a_{ex}$. Studying 2D sheets, the binding energy is given in [56] using the parabolic approximation for the kinetic energy and the Rytova-Keldysh potential in Eq. (10). Here, instead, with the Dirac band dispersion in the kinetic energy of Eq. (7), we find a modified kinetic energy

$$E_{kin}(\lambda) = \frac{E_g}{2} \frac{x}{1-x} \left\{ 1 - \frac{x}{2} \left[ \frac{\theta(1-x)}{\sqrt{1-x}} \ln\left(\frac{1+\sqrt{1-x}}{1-\sqrt{1-x}}\right) + 2\frac{\theta(x-1)}{\sqrt{x-1}} \arctan\sqrt{x-1} \right] \right\}, \tag{13}$$

and, therefore, a binding energy

$$E_b(\lambda) = -E_{\text{kin}}(\lambda) + \frac{e^2}{\bar{\epsilon}\rho_0}\frac{1}{\sqrt{1+\beta^2}}\left[\frac{\ln\left(\sqrt{1+\beta^2}+\beta\right)+\ln\left(\sqrt{1+\frac{1}{\beta^2}}+\frac{1}{\beta}\right)}{1+\beta^2} - \frac{1-\beta}{\sqrt{1+\beta^2}}\right]. \quad (14)$$

The dependence on the dimensionless variational parameter $\lambda$ is through $x = \frac{8R_{ex}}{E_g}\lambda^2$ and $\beta = \frac{a_{ex}}{4\rho_0}\frac{1}{\lambda}$. Here, the parameters $R_{ex} = R_H\frac{\mu}{m\bar{\epsilon}^2}$ and $a_{ex} = a_B\bar{\epsilon}\frac{m}{\mu}$ of a 3D Wannier-Mott model with the hydrogen Rydberg energy $R_H = 13.605$ eV and the atomic Bohr radius $a_B = 0.529$ Å have been formally introduced.

The first contribution, the negative kinetic energy, vanishes for $x = 0$, i.e., flat Dirac bands with $v_F \to 0$ and $\mu \to \infty$. In the opposite limit $x \to \infty$, i.e., $E_g \to 0$ and $\mu \to 0$, the kinetic energy becomes $\frac{\pi}{4}\sqrt{x}E_g$, the value of pure linear bands. The value $\frac{1}{2}xE_g$, following within the EMA of the kinetic energy operator in Eq. (8), cannot be obtained from Eq. (13) because the limits $x \to \infty$ with $v_F \to 0$ or $E_g \to \infty$ cannot be interchanged with the infinite integral of the matrix element calculation. The second contribution, the negative potential energy, also shows two characteristic limits. For $\beta \ll 1$, i.e., in the large polarizability/small excitonic radius limit, it shows a logarithmic behavior $-2R_{ex}\frac{a_{ex}}{\rho_0}[\ln(\beta/2)+1]$. In the opposite limit $\beta \gg 1$, i.e., vanishing 2D polarizability/large exciton radius, one finds $8R_{ex}\lambda$, the Coulomb result of the 2D hydrogen atom.

## 3 Exciton binding versus gap

Within the single-particle approach, the optical absorption edge is given by the SOC-induced fundamental gap $E_g$ and subsequent interband transitions (see Fig. SM1 in the Supplemental Material). It raises the question about what happens after inclusion of the excitonic effects. In a conventional semiconductor with $E_g > E_b$ the appearance of excitonic bound states is expected. In the studied small gap systems, the Xenes, with reduced screening due to their low dimensionality, the occurrence of bound excitons with $E_b > E_g$, i.e., the formation of a spontaneously formed new electronic ground state, the EI phase, has to be investigated in a more rigorous way.

### 3.1 Lowest bound exciton

For freestanding Xenes with $\bar{\epsilon} = 1$ the binding energies $E_b$ and the excitonic radii $r_{ex}$ are listed in Table 2. They are computed by means of the variational procedure of Eq. (13) and $r_{ex} = a_{ex}/(2\lambda)$, using the three different types of static electronic polarizabilities $\alpha_{2D}(\text{bulk})$, $\alpha_{2D}(\text{model})$ and $\alpha_{2D}(\text{DFT})$ given in Table 1. The resulting values are compared with those obtained replacing the kinetic energy operator in Eq. (7) by that in EMA from Eq. (8). Independent of the approximation used for the kinetic energy of the internal exciton motion and the actual screening described by $\alpha_{2D}$, common chemical trends are visible. In general, the exciton binding energies increase along the series Si→Ge→Sn→Pb in a similar way as the fundamental gap energy $E_g$. The only exception occurs for Dirac-like kinetic energies and the use of bulk screening $\alpha_{2D}(\text{bulk})$, where the opposite trend of $\alpha_{2D}$ rules that of the binding energies $E_b$, apart from plumbene, whose corresponding bulk is metallic. Using $\alpha_{2D}(\text{bulk})$, the binding energies $E_b$ exceed the gap values $E_g$ because of the extremely small screening. As a consequence, within the $\alpha_{2D}(\text{bulk})$ approximation EI phases are predicted for silicene, germanene, and stanene. In the cases of the screening calculated using $\alpha_{2D}(\text{model})$ and $\alpha_{2D}(\text{DFT})$, instead, the binding energies $E_b$ are smaller, and close to the gap values $E_g$ reported in Table 1. This

Table 2: Excitonic parameters, binding energy $E_b$ and characteristic radius $r_{ex}$, of a real or fictitious lowest bound exciton from the variational procedure described in Eq. (14). Two different approximations of the kinetic energy, Eqs. (7) and (8), and three different screenings of the electron-hole attraction, Eq. (10), expressed by the static electronic polarizabilities $\alpha_{2D}$ in Table 1, are applied. Binding energies $E_b > E_g$ are indicated in red.

| kinetic energy | Xene | silicene | germanene | stanene | plumbene | screening |
|---|---|---|---|---|---|---|
| Dirac-band | $E_b$ (meV) | 406 | 298 | 238 | 0 | $\alpha_{2D}$(bulk) |
| | | 1.5 | 22.6 | 76.7 | 500 | $\alpha_{2D}$(model) |
| | | 1.2 | 19.9 | 69.6 | 423 | $\alpha_{2D}$(DFT) |
| | $r_{exc}$ (nm) | 1.3 | 1.9 | 2.5 | $\infty$ | $\alpha_{2D}$(bulk) |
| | | 414 | 26.2 | 7.3 | 1.1 | $\alpha_{2D}$(model) |
| | | 495 | 29.3 | 8.0 | 1.3 | $\alpha_{2D}$(DFT) |
| EMA | $E_b$ (meV) | 11.9 | 104 | 166 | 0 | $\alpha_{2D}$(bulk) |
| | | 1.3 | 19.9 | 67.7 | 441 | $\alpha_{2D}$(model) |
| | | 1.1 | 17.8 | 62.1 | 381 | $\alpha_{2D}$(DFT) |
| | $r_{exc}$ (nm) | 121 | 12.2 | 5.9 | $\infty$ | $\alpha_{2D}$(bulk) |
| | | 613 | 38.2 | 10.7 | 1.6 | $\alpha_{2D}$(model) |
| | | 689 | 41.1 | 11.3 | 1.8 | $\alpha_{2D}$(DFT) |

is due to the fact that the 2D materials here considered have a gap ruled by SOC. Hence, the appearance or disappearance of an EI phase is difficult to predict. The $E_b$ trend is also similar to the trend of the 2D hydrogen atoms with $E_b^{2DH} = 4R_H \mu/m$, 14 (Si), 214 (Ge), 836 (Sn) and 906 (Pb) meV. However, the values of 2D hydrogen atoms are much larger, because they define the upper limit, with a screened interaction $\hat{W}(\mathbf{x}) = -\frac{e^2}{\bar{\epsilon}\rho}$, of the Rytova-Keldysh potential in Eq. (10). The exciton radii $r_{ex}$ follow reversed chemical trends, in accordance with the values of the 2D hydrogen, $r_{ex}^{2DH} = a_B m/(2\mu)$, 106 (Si), 7 (Ge), 2 (Sn) and 0.2 (Pb) nm, as well as with the characteristic screening radii $\rho_0 = 2\pi\alpha_{2D}$, 138 (Si), 94 (Ge), 28 (Sn) and 5 (Pb) nm using $\alpha_{2D}$(DFT).

The general trends of the exciton binding parameters $E_b$ and $r_{ex}$ are plotted in Fig. 1 and Fig. 2, respectively, versus the screening radius $\rho_0 = 2\pi\alpha_{2D}$, as obtained within the variational approach with Dirac kinetic energy (Eq. (14), red lines) and EMA (blue line). The results are normalized to the parameters $R_{ex}$ and $a_{ex}$ of the corresponding Wannier-Mott excitons. Notice that these normalization factors are material-dependent through $\mu$. Therefore, the normalized exciton binding parameters $E_b/R_{ex}$ and $r_{ex}/a_{ex}$ show an opposite trend as a function of the specific Xene, compared with the trend of their absolute values $E_b$ and $r_{ex}$. Concerning the exciton binding energy, Fig. 1 shows that, while the blue curve (EMA kinetic energy) tends to the finite value $E_b/R_{ex} = 4$ in the 2D H limit, the red ones (Dirac kinetic energy) give a diverging value of $E_b/R_{ex}$ in the same limit.

The different kinetic energy approximations, Dirac or EMA, to describe $E_b$ and $r_{ex}$ have only a relatively weak influence on the exciton parameters. In other words, for a given polarizability evaluation method, and for a given Xene, $E_b$ and $r_{ex}$ are not very dependent on the description of the interband dispersion. The binding energies $E_b$ within the Dirac approximation for the kinetic energy (exciton radii $r_{ex}$) are only slightly larger (smaller) than the values within the EMA. Also the dependence on $E_g/R_{ex} \sim v_F^2$ for the four Xenes is of minor influence in the Dirac kinetic energy approach, as shown in Figs. 1 and 2, where two values for $E_g/R_{ex}$ have been used. The two red lines, calculated for silicene ($E_g/R_{ex}$ =0.47, solid line) and for plumbene ($E_g/R_{ex}$ =0.34, dashed line) almost overlap.

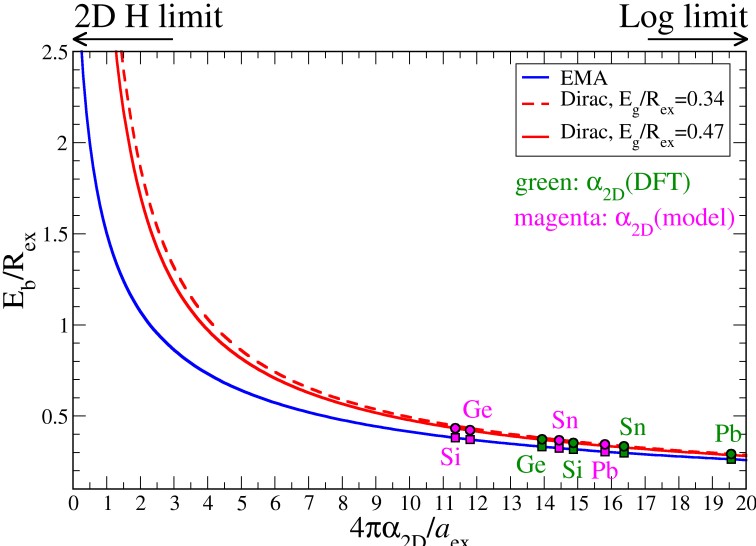

Figure 1: Exciton binding energy $E_b$ measured in units of the Wannier-Mott exciton parameter $R_{ex}$, versus twice the normalized screening radius $\rho_0 = 2\pi\alpha_{2D}$. Variational results with the potential energy as the second term in Eq. (14) and the kinetic energy in EMA of Eq. (8) are displayed in blue line, while those with the non-parabolic kinetic energy (Eq. (7)) appear as red lines. In the latter case the parameters also depend on the gap energy. This is illustrated by variation of $E_g/R_{ex} = 0.34$ (dashed red line, plumbene) to 0.47 (solid red line, silicene). The specific values of the exciton binding parameters obtained for the four Xenes are highlighted for the two cases $\alpha_{2D}(DFT)$, in green, and $\alpha_{2D}(model)$, in magenta. The labels appearing above the plot refer to the two limit cases of the screened interaction: unscreened hydrogen model or logarithmic behavior. While the blue curve (EMA kinetic energy) tends to the finite value $E_b/R_{ex} = 4$ in the 2D hydrogen limit, the red ones (Dirac kinetic energy) give a diverging value of $E_b/R_{ex}$ in the same limit.

In the absence of external electric fields, silicene, germanene and stanene represent topological insulators with $\mathbb{Z}_2 = 1$ [21–25], while we found a trivial insulator with $\mathbb{Z}_2 = 0$ for plumbene, in agreement with other studies [41, 44, 45, 57]. In contrast to the principal chemical trends in exciton binding, the relation of $E_b$ to the fundamental gap $E_g$ and, therefore, the prediction of an EI with $E_b/E_g > 1$ or normal semiconductor with $E_b/E_g < 1$ significantly depends on the screening of the electron-hole attraction $\hat{W}$ (Eq. (10)). Apart from plumbene, the bulk-like screening by $\alpha_{2D}(bulk)$ in Table 1 is much smaller than that characterized by $\alpha_{2D}(model)$ and $\alpha_{2D}(DFT)$, giving large binding energies. As a consequence, it holds $E_b/E_g > 1$ when $\alpha_{2D}(bulk)$ is used, indicating the existence of the EI phase, in complete agreement with the findings of Brunetti et al. [28]. When more refined approximations of the screening ($\alpha_{2D}(DFT)$ or $\alpha_{2D}(model)$) are used, instead, no excitonic insulator phase is predicted. In the case of plumbene, the opposite behavior is found in Table 2, at least for the Dirac kinetic energy approximation in Eq. (7): plumbene is predicted to be an excitonic insulator when using $\alpha_{2D}(model)$ and a trivial insulator when using $\alpha_{2D}(DFT)$. The sensitivity of the appearance of the EI phase on the exact magnitude of the electronic polarizability $\alpha_{2D}$ suggests also strong influence of an additional screening, if the Xene sheet is embedded in dielectrics with $\bar{\epsilon} > 1$. For instance, encapsulation of the Xene sheet by hexagonal BN layers with an averaged static electronic dielectric constant $\bar{\epsilon} \approx 4.3$ [58] leads to a further reduction

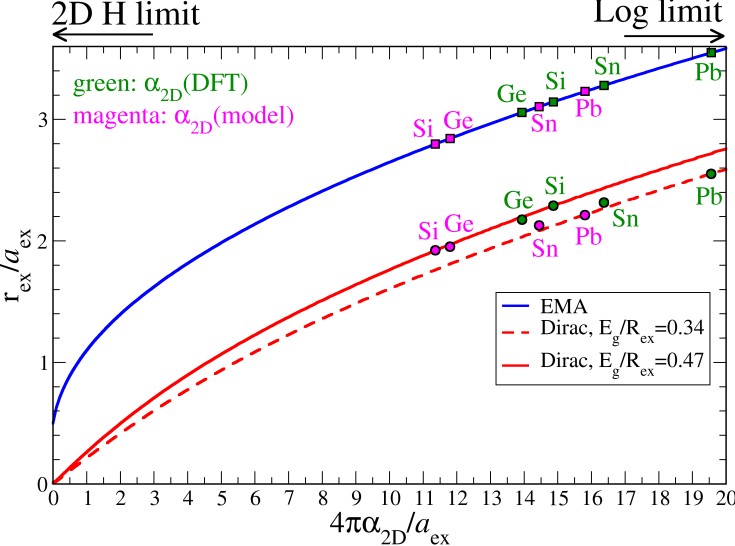

Figure 2: Exciton radius $r_{ex}$ measured in units of the Wannier-Mott exciton parameter $a_{ex}$, versus twice the normalized screening radius $\rho_0 = 2\pi\alpha_{2D}$. Variational results with the potential energy as the second term in Eq. (14) and the kinetic energy in EMA (Eq. (8)) are displayed in blue line, while those with the non-parabolic kinetic energy (Eq. (7)) appear as red lines. In the latter case the parameters also depend on the gap energy. This is illustrated by variation of $E_g/R_{ex} = 0.34$ (dashed red line, plumbene) to 0.47 (solid red line, silicene). The specific values of the exciton binding parameters obtained for the four Xenes are highlighted for the two cases $\alpha_{2D}$(DFT), in green, and $\alpha_{2D}$(model), in magenta. The labels appearing above the plot refer to the two limit cases of the screened interaction: unscreened hydrogen model or logarithmic behavior.

of the electron-hole attraction. Consequently, the dielectric embedment of a Xene sheet tends to reduce the probability to explore the excitonic insulator phase.

The influence of the screening by the electron ensemble, i.e., the electronic polarizability is more clearly represented in Fig. 3 by plotting, in units of the SOC-induced gap $E_g$, the binding energy (Table 2) versus the unscreened 2D hydrogen atom energy (obtained from the parameters given in Table 1). The latter quantity refers to the binding energy of an unscreened 2D hydrogen atom with a kinetic energy in EMA and the Coulomb attraction given as a bare Coulomb potential in 2D. The actual band dispersion used to model the kinetic energy of the internal exciton motion plays a minor role. More important is the modeling of the screening by $\alpha_{2D}$(bulk), $\alpha_{2D}$(model) or $\alpha_{2D}$(DFT). Thereby, apart from plumbene, the bulk-like screening with relatively small $\alpha_{2D}$(bulk) values clearly suggests that the Xenes are EIs. Apart from plumbene and $\alpha_{2D}$(model), where also an EI situation appears, the majority of other $E_b/E_g$ ratios when $\alpha_{2D}$(model) or $\alpha_{2D}$(DFT) are used, are smaller than 1. However, all the values are close to the phase boundary (dashed horizontal line in Fig. 3), between the EI phase and the normal semiconductor phase. Summarizing, Fig. 3 shows a strong influence of screening and its description. Therefore, general predictions are difficult.

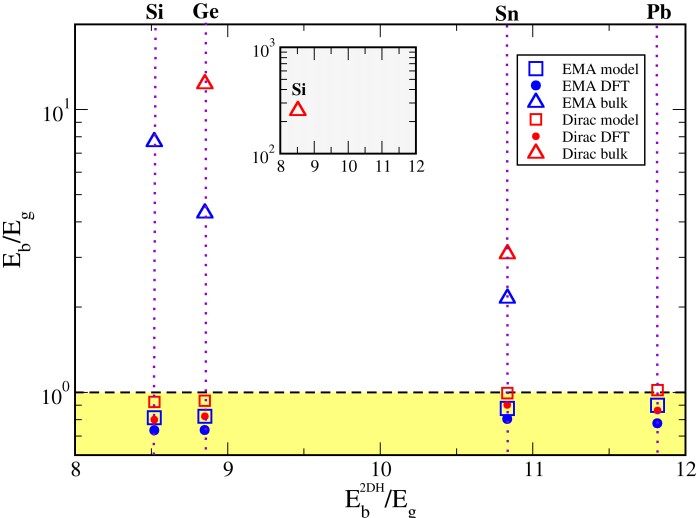

Figure 3: Ratio of exciton binding energy $E_b$ to the gap $E_g$ versus the ratio of binding within the 2D hydrogen atom model $E_b^{\mathrm{2DH}}$ to the gap $E_g$. The symbols represent different electronic polarizabilities: triangles $\hat{=} \alpha_{\mathrm{2D}}$(bulk), squares $\hat{=} \alpha_{\mathrm{2D}}$(model), and dots $\hat{=} \alpha_{\mathrm{2D}}$(DFT). The red (blue) symbols show values obtained with the kinetic energy in Dirac approximation (EMA). The dashed horizontal line defines the boundary between the trivial insulator phase (yellow region) and the EI phase.

## 3.2 Application of a vertical electric field

An external vertical electric field drastically changes the band structure, Eq.(1), of the Xenes around a $K$ or $K'$ point. This holds especially for the direct fundamental gap $E_g \rightarrow E_g(U)$, Eq. (2), at the BZ boundary points. As a consequence, a significant modification of the static electronic polarizability is expected. According to the approximate formula in Eq. (12), its general influence can be written as

$$\alpha_{\mathrm{2D}}(U) = \alpha_{\mathrm{2D}} \frac{E_g}{E_g(U)} . \tag{15}$$

The modified quantities $E_g(U)$ and $\alpha_{\mathrm{2D}}(U)$ allow to calculate the field influence on the excitonic binding according to expression in Eq. (13) (Dirac) or in the EMA approximation.

More in detail, the influence of the bias $U$ on the exciton binding $E_b = E_b(U)$ is demonstrated in Figs. 4, 5 and in Fig. SM2 (see Supplemental Material) in comparison with the actual direct gap $E_g(U)$. Figures 4 and 5 illustrate the exciton binding using the complete massive Dirac band dispersion around $K$ and $K'$ expressed by the kinetic energy in Eq. (7). The two figures only differ with respect to the use of the 2D screening, $\alpha_{\mathrm{2D}}$(model) in Fig. 4 and $\alpha_{\mathrm{2D}}$(DFT) in Fig. 5. In order to identify the existence of an EI phase, the field-dependent fundamental gap is used in the kinetic energy (Eq. (7)) of the internal exciton motion. $E_g(U)$ is also displayed (black lines). It shows the well-known linear variation with a zero at the value of the critical field strength [17,21,23,24,32]. The region of the decreasing gap up to zero corresponds to the TI phase of the Xene with a topological invariant $\mathbb{Z}_2 = 1$ [17,21], and is marked in yellow in Figs. 4 and 5. The region with increasingly larger field and increasing gap $E_g(U)$ describes the trivial phase of the Xenes with the topological invariant $\mathbb{Z}_2 = 0$ [17,24]. Thereby the topological invariant $\mathbb{Z}_2$ is calculated in different ways for the centrosymmetric unbiased Xenes [59] and the non-centrosymmetric systems if an electric field is applied [43].

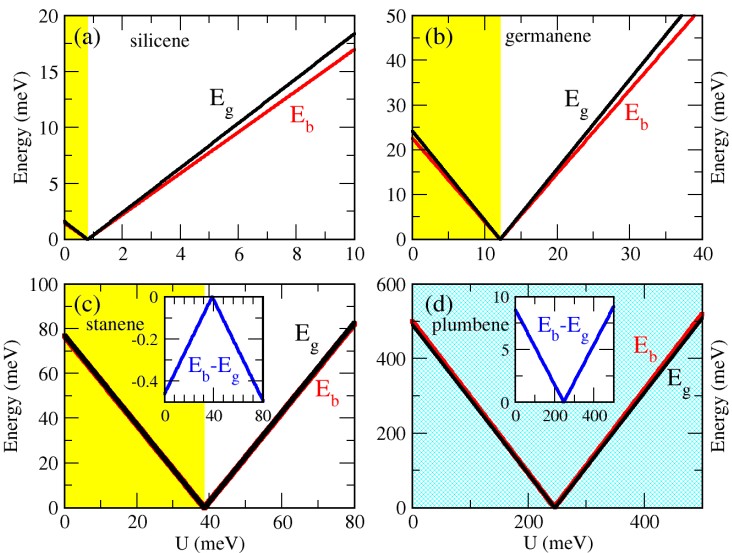

Figure 4: Exciton binding energy $E_b(U)$ (red lines), using the Dirac approximation for the kinetic energy, and direct fundamental gap $E_g(U)$ (black lines) at $K/K'$ as a function of an applied potential energy difference $U$. The screening by $\alpha_{2D}$(model), Eq. (15), is used. The occurrence of an excitonic insulator phase is indicated by a cyan background, while the topological insulator region is shown by a yellow background.

The exciton binding energies $E_b(U)$ also show a linear behavior with $U$, with a vanishing value at the critical value $U_{\text{crit}} = E_g/2$ because of the infinite screening due to $\lim_{U \to U_{\text{crit}}} \alpha_{2D}(U) \to \infty$, according to Eq. (15). Independently on the used electronic polarizability approach, $\alpha_{2D}$(model) or $\alpha_{2D}$(DFT), seven out of eight panels in Figs. 4 and 5 indicate $E_g(U) > E_b(U)$, i.e., the non-existence of an EI phase, but instead a normal semiconductor with excitonic bound states below the absorption edge $E_g(U)$. Only the panel for plumbene in Fig. 4 indicates $E_b(U) > E_g(U)$, i.e., the possible existence of an excitonic insulator phase. However, as illustrated in the figures, the situation is not fully recognizable because of the closeness of $E_b(U)$ and $E_g(U)$. A more detailed analysis of the data in Fig. 4 (see insets) shows that $E_b(U) \stackrel{<}{\sim} E_g(U)$ with a minor difference of few meV. However this conclusion is rather fragile as indicated by the difference $E_b(U) - E_g(U)$ in the insets of the stanene and plumbene panels of Fig. 4. The positive difference in the plumbene case remains small with a variation between 0 and 10 meV, and the negative difference $E_b(U) - E_g(U)$ in the stanene case is even smaller.

The influence of the kinetic energy operator on the field-modified excitonic binding is illustrated in Fig. SM2 (see Supplemental Material), applying the EMA (Eq. (8)) but keeping the screening by $\alpha_{2D}$(DFT) as in Fig. 5. For comparison, the same linear variations $E_g(U)$ of the fundamental gap are displayed. The modification of the kinetic energy of the internal exciton motion from Dirac (Eq. (7)) to EMA approximation (Eq. (8)) leads to small changes of the $E_b(U)$ curves. Still $E_b(U_{\text{crit}}) = 0$ is conserved. This tendency widely disagrees with the findings of Brunetti et al. [28], who also applied the EMA but have taken a field-independent screening ruled by $\alpha_{2D}$(bulk). As an example, we show in Fig. 6 the effect of the different approximations on the binding energy of stanene. Because of the constant screening, the field-modified

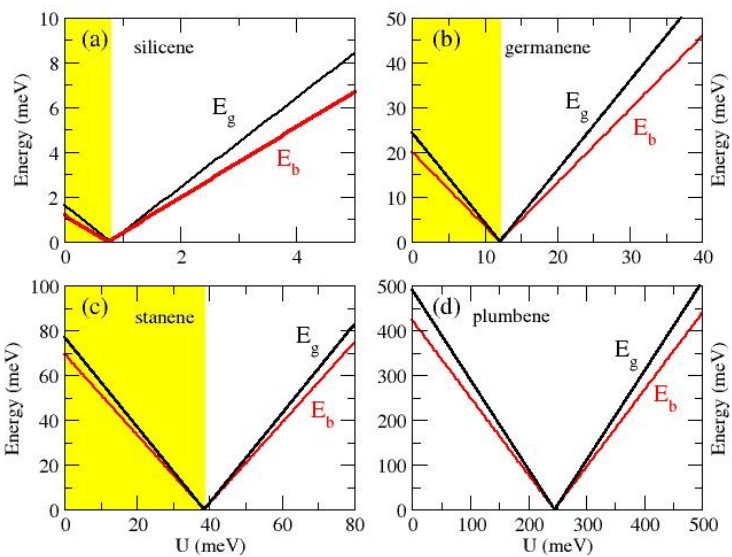

Figure 5: As Fig. 4 but for $\alpha_{2D}$(DFT). The yellow background indicates the TI phase. No EI phase is found.

behavior away from the critical bias region is strongly nonlinear (blue curve in Fig. 6). The binding energy, both in the Dirac (green curve) and in the EMA (blue curve) approximation, is always larger than the gap $E_g$, pointing towards an excitonic insulator phase. The opposite conclusion is instead reached when using EMA and a field-dependent polarizability $\alpha_{2D}$(DFT) (red curve).

The observation of the EI phase around $U_{crit}$, when a field-independent screening $\alpha_{2D}$(bulk) is used, is also in complete contrast to the findings in Figs. 4c and 5c, calculated in the framework of the kinetic energy in Dirac approximation (Eq. (7)). The correct description of the screening is hence of primary importance in the quest for a EI phase, whereas the treatment of the kinetic energy, Dirac instead of EMA, is of secondary importance, and gives minor differences in the resulting binding energies. The main reason for the small discrepancy between results for the different kinetic energies, Eqs. (7) and (8), is understandable if we consider the consequences of the gap variation $E_g(U)$ with $U$. Near $U = U_{crit}$ the gap vanishes, $E_g(U_{crit}) = 0$. This means that in the vicinity of $U_{crit}$ the EMA is not valid anymore, since the bands become linear and the kinetic energy is no longer given by Eq. (8), but is $\hat{T}(\mathbf{x}) = 2i\hbar v_f \boldsymbol{\nabla_x}$, i.e., linear in the momentum operator $\mathbf{p} = -i\hbar\boldsymbol{\nabla_x}$, and Dirac massless fermions appear. But it is worth to stress once more that the approximations used for the screening are of overwhelming importance for a correct determination of the excitonic binding energy. The use of a bulk-derived $\alpha_{2D}$(bulk) predicts the existence of the EI phase in silicene, germanene and stanene, in contrast with the results obtained with an analytical ($\alpha_{2D}$(model)) or a numerical ($\alpha_{2D}$(DFT)) evaluation of the two-dimensional polarizability.

## 3.3 Comparison with *ab initio* many-body perturbation theory: The case of stanene

In order to shed light on the possible existence of an EI phase for Xenes, *ab initio* calculations based on Many-Body perturbation theory have been performed and results compared

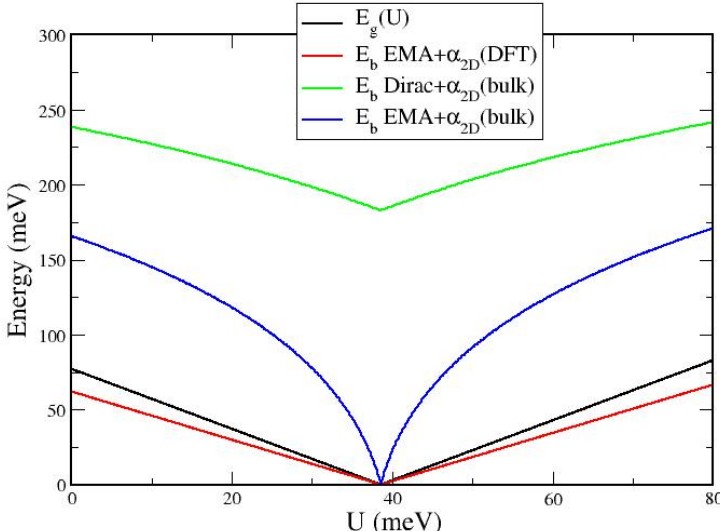

Figure 6: Exciton binding energy $E_b(U)$ for stanene as a function of the applied potential energy difference $U$, calculated at different levels of approximations. Red lines: within the EMA approximation for the kinetic energy and using the numerical *ab initio* $\alpha_{2D}(\text{DFT})(U)$ for the screening. Blue: EMA approximation, but with a bulk-derived screening $\alpha_{2D}(\text{bulk})=\epsilon\delta/4\pi$ independent on $U$. Green: within the Dirac bands approximation, with a bulk-derived screening $\alpha_{2D}(\text{bulk})$ independent on $U$. In black the direct gap $E_g(U)$ at $K/K'$ is also reported.

with the those obtained by modeling the kinetic energy and the screening. GW plus BSE approaches [46, 60] represent by now the state of the art, and most accurate way, to evaluate the electronic and the optical gaps of materials. The difference between the electronic and optical gaps gives the exciton binding energy, which is the quantity needed to understand if an EI phase may occur or not. Here, we present GW+BSE results for stanene and compare them with those obtained within the EMA and the Dirac form of the kinetic energy, using the numerical $\alpha_{2D}(\text{DFT})$, the analytical $\alpha_{2D}(\text{model})$ and the bulk-derived $\alpha_{2D}(\text{bulk})$ screen. Many-body calculations are extremely heavy on these group-IV Xenes because of the need of a dense **k**-points mesh near the Dirac point. This is especially true for silicene and germanene, because of the very small SOC gap. We chose hence stanene, whose 77.2 meV DFT gap makes it a good candidate to achieve converged results with an acceptable computational effort. GW and BSE calculations were performed using the Yambo code [61, 62]. Spin-orbit corrections were included. The screening $W$ and the correlation part of the self-energy $\Sigma_c$ were calculated using 500 empty bands, 72×72×1 **k**-points and a cutoff of 8 Hartree. As it is well known, the analysis of 2D systems within periodic boundary requires particular care in order to avoid the spurious interaction between stanene monolayers in neighboring supercells. At the DFT level it is sufficient to build a supercell with a *vertical* amount of vacuum space which preserves the ground state density of the isolated layer; in the present case a distance of 12 Å between the sheets replica is adequate to this scope. Instead, at the GW and BSE level the convergence of the Coulomb integral requires an explicit cutoff procedure which cuts the long-range part of the interaction by modifying the expression for the Coulomb operator in reciprocal space [63, 64] . Calculations performed in this paper make usage of this technique, as implemented in the Yambo code, where also Monte Carlo integration method is introduced to deal

with the long-wavelength limit of the 2D screening [65]. For an investigation of the influence of the cutoff procedures on optical properties and screening see also [66].

The BSE pair excitation energy was calculated using a cutoff of 3 Hartree, 72×72×1 **k**-points, two empty bands and two valence bands. Convergence tests with 18×18×1 and 30×30×1 **k**-points meshes were performed. The resulting GW electronic gap of stanene amounts to 176 meV, with an exciton binding energy of 77 meV, in excellent agreement with the values 76.7 meV and 69.6 meV obtained within the Dirac kinetic energy approximation, using $\alpha_{2D}$(model) and $\alpha_{2D}$(DFT), respectively. On the other hand, our Many-Body GW+BSE calculations demonstrate that the values of E$_b$ (238 and 166 meV in Dirac and EMA approximation, respectively), obtained approximating the screening by $\alpha_{2D}$(bulk), heavily overestimate the true stanene excitonic binding energy, thus highlighting the importance of an appropriate modeling of the screen in two-dimensional materials. Remarkably, being the GW gap (176 meV) larger than the BSE exciton binding energy (77 meV), we conclude that stanene is not an excitonic insulator, confirming the results obtained with the Dirac and EMA kinetic energies in conjunction with the *ab initio* $\alpha_{2D}$(DFT) and $\alpha_{2D}$(model), thus validating the models used. Finally, it is worth to point out that the analytical determination of $\alpha_{2D}$(model) through Eq. (12) needs just the calculation of the electronic gap. The simplicity of this approach as compared with the calculation of $\alpha_{2D}$(DFT) is especially appreciable when small (∼meV) Dirac gaps appear, which cause a very slow convergence of the low-energy part of optical spectra and the need of using thousands of **k**-points near the gap.

## 4 Summary and conclusions

The possibility of the occurrence of the excitonic insulator phase has been investigated by a variational method for the binding energy and the radius of the lowest-energy bound exciton in the Xenes silicene, germanene, stanene and plumbene. Thereby, two different approximations of the kinetic energy of the internal exciton motion, one based on Dirac bands modified by spin-orbit coupling and one corresponding to the effective-mass approximation of those bands, have been used. The screening of the electron-hole attraction in the freestanding Xenes has been characterized by a static electronic polarizability and the resulting Rytova-Keldysh potential, which can be derived assuming 2D electronic systems or for 3D crystals in the limit of vanishing thickness. Consequently, we applied two different classes of approaches to the determination of the 2D electronic polarizabilities: a bulk-like one $\alpha_{2D}$(bulk) resulting within a quantum-well treatment of an isolated Xene sheet, and a direct calculation of the 2D electronic polarizability within two approaches, an analytical relation $\alpha_{2D}$(model) to the inverse fundamental gap $1/E_g$, and an *ab initio* calculation of $\alpha_{2D}$(DFT) within the independent-particle approximation. The resulting exciton binding energies $E_b$ have been compared with the absorption edge at the fundamental gap $E_g$. For $E_b < E_g$, the Xene should be a normal semiconductor with bound excitons redshifted to the absorption edge. For $E_b > E_g$, an excitonic insulator phase is predicted for the Xene sheet.

We found a minor influence of the chosen kinetic energy on the exciton binding. The influence of the chosen screening of the electron-hole attraction is much stronger. Opposite results have been observed for $\alpha_{2D}$(bulk), ruled by the bulk *sp*-gap, and the two other screening approaches $\alpha_{2D}$(model) and $\alpha_{2D}$(DFT), ruled by the SOC-induced sheet gap: opposite chemical trends are observed, but also significantly different absolute values. Chemical trends and absolute values of $\alpha_{2D}$(bulk) seem to indicate that the description of screening by bulk dielectric constants is not valid for Xenes with a SOC-induced gap. As a consequence, the use of $\alpha_{2D}$(bulk) tends to favor the excitonic insulator phase in agreement with previous predictions, while the stronger screenings by $\alpha_{2D}$(model) and $\alpha_{2D}$(DFT) tend to result in the trivial

insulator phase with $E_b < E_g$.

Despite the strong modification of $E_b$ and $E_g$ by a vertical electric field realized by bias voltage $U$, the general trend $E_b < E_g$ is conserved when the screening is described by $\alpha_{2D}(\text{model})(U)$ or $\alpha_{2D}(\text{DFT})(U)$. However, totally different results are observed when using a $U$-independent $\alpha_{2D}(\text{bulk})$ as in [27,28] at any voltage, predicting an excitonic insulator phase. Moreover, around the critical value $U_{\text{crit}} = E_g/2$ separating the topological and the trivial phases (with the exception of plumbene), the EMA and Dirac descriptions give very different binding energy, zero in the first case, a finite value in the second one. Since EMA describes parabolic bands, it is expected to be a poor approximation to describe the bound excitons for vanishing fundamental gaps, where the bands are linear. Therefore, the model resulting from the use of the Dirac-like kinetic energy, introduced in this work, provides more reliable predictions about the occurrence of the EI phase. Consequently, we clearly favor a description of the low-energy excitons in Xenes, which takes the Dirac-like character of electrons and holes into account as well as a Coulomb potential screening by static electronic polarizabilities, which are ruled by the electronic band structure near the $K/K'$ points.

Our hypothesis is corroborated by many-body perturbation theory calculations performed within the GW approximation and solving the Bethe-Salpeter equation. We find for stanene an excitonic binding energy of 77 meV, in excellent agreement with the value 69.6 meV found using the 2D excitonic model based on the Dirac bands kinetic energy and on the *ab initio* DFT screening. The approximate analytical description of the 2D polarizability by the inverse gap comes with 76.7 meV, even closer to the GW/BSE binding energy. Our many-body results thus confirm the absence of an excitonic insulator phase in stanene and validate the models used for $\alpha_{2D}$ by taking the true band structure of the atomic sheets into account. Last but not least, we demonstrate that a simple analytical approximation for the screening, namely $\alpha_{2D}(\text{model})$ based on Eq. (12), represents a fast, easy and reliable way to calculate the excitonic binding energy of 2D Dirac materials.

## Acknowledgments

We thank Dr. Daniele Varsano for enlightening discussions.

**Funding information** O.P. acknowledges financial funding from the INFN project TIME2QUEST and from PRIN 2020 "PHOTO". F. B. acknowledges financial support from INFN Tor Vergata. CPU time was granted by CINECA HPC center.

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
