# Peer review of "Transitions in Xenes between excitonic, topological and trivial insulator phases: influence of screening, band dispersion and external electric field"

_SciPost Physics, doi:SciPost Phys. 15, 025 (2023)_

## Round 1 · Referee Report · Anonymous (Referee 1) · 2023-3-20

Strengths

1) topic is important 2) approach is pertinent and (for the validation) state of the art

Weaknesses

1) presentation of results

Report

In this article, the authors present a study to evaluate the
exciton properties of xenes. In particular, they study the
influence of the e-h screening (and band dispersion and the
role of an external electric field) in discriminating between
an insulator and a candidate topological insulator.

The analysis of the screening is of great importance in the
description of optical properties. But in these systems
(2D, small band-gap, spin-orbit coupling) it becomes a very
cumbersome part of the calculation if the ab initio approach
is uniquely considered. The proposition of models is then more
than welcome. For this reason, I believe the results shown
in the article deserve publication.
I find however the presentation unclear. I believe a little effort
would permit the article to be better structured, with a clear
discussion of the results, a summary of the different models, etc.

The article should be published only after this work has been
carried out.

Requested changes

1) the discussion of the different models, their physical meaning, the advantages and the limits should been highlighted, within a summary. I would expect much clearer indication where a model makes sense, where we should expect problems, etc. All this is sort of diluted all along the text and not very clear.

2) at the beginning of II.B, the authors mention the ingredients of the problem in BSE: electron(hole) energy bands, statically screened Coulomb interaction W and bare repulsive e-h exchange. I do not find the latter term in Eq.3, while I find the first two terms. Is it embedded in W? Is it excluded from the model?

3) In the ab initio calculation, it is mentioned a cutoff in the Coulomb interaction. I understand this as a mean to overcome the problem of a supercell replica. But I would like more details, for the use of a cutoff introduce a parameter. The problem of calculating surfaces or slabs in periodic boundary conditions is a long standing one, often treated very unsatisfactorily. Recently an article shed a bit of light in this matter: Phys. Rev. B 106, 035431. Maybe it relates.

4) Slightly related to point 1 above: the ab initio 'validation' comes only from the case of stanene. Considering the variation of results of the models and the comparison with only one case, I would tone down the validation. What is the amount of calculation here involved (in terms of computational resources) and how much would be required to do an extra case ? Here, I want to stress that I totally am in favour of the objective of the article: to offer a much cheaper alternative (by using models) to the ab initio brute force approach of exciton binding energy. But a validation has to be meaningful.

  • validity: high
  • significance: high
  • originality: high
  • clarity: low
  • formatting: good
  • grammar: good

Author:  Marco D'Alessandro  on 2023-04-28  [id 3624]

(in reply to Report 1 on 2023-03-20)
Category:
answer to question
reply to objection

**The referee writes:**
_The analysis of the screening is of great importance in the
description of optical properties. But in these systems
(2D, small band-gap, spin-orbit coupling) it becomes a very
cumbersome part of the calculation if the ab initio approach
is uniquely considered. The proposition of models is then more
than welcome. For this reason, I believe the results shown
in the article deserve publication_

**Our response:**

We thank the reviewer for his/her clear and positive characterization of our work and the recommendation of publication of our manuscript.

**The referee writes:**
_I find however the presentation unclear. I believe a little effort
would permit the article to be better structured, with a clear
discussion of the results, a summary of the different models, etc.
The article should be published only after this work has been
carried out._

**Our response:**
We thank the reviewer for his/her suggestions and have correspondingly improved the structure, discussion and summary in the resubmitted manuscript.

__Requested changes__

**The referee writes:**
_1)The discussion of the different models, their physical meaning,
the advantages and the limits should been highlighted, within a
summary. I would expect much clearer indication where a model
makes sense, where we should expect problems, etc.
All this is sort of diluted all along the text and not very clear._

**Our response:**
We thank the reviewer for this suggestion. The different models are discussed in the new text under varying aspects.

**The referee writes:**
_2)at the beginning of II.B, the authors mention the ingredients
of the problem in BSE: electron(hole) energy bands, statically screened
Coulomb interaction W and bare repulsive e-h exchange. I do not
find the latter term in Eq.3, while I find the first two terms.
Is it embedded in W? Is it excluded from the model?_

**Our response:**
In the investigated limit of the description of the electron-hole interaction within a two-band model and vanishing translation of the excitons the electron-hole exchange is vanishing as well known from the description of the Wannier-Mott excitons. We mention this fact now in the text of the resubmitted manuscript.

**The referee writes:**
_3) In the ab initio calculation, it is mentioned a cutoff in the
Coulomb interaction. I understand this as a mean to overcome the
problem of a supercell replica. But I would like more details, for
the use of a cutoff introduce a parameter. The problem of calculating
surfaces or slabs in periodic boundary conditions is a long standing one,
often treated very unsatisfactorily. Recently an article shed a bit
of light in this matter: Phys. Rev. B 106, 035431. Maybe it relates._

**Our response:**
The treatment of the artificial Coulomb interaction between different slabs in the supercell arrangement is now discussed describing the ab initio treatment of quasiparticle GW and excitonic BSE. Moreover, details and references are given. Referring to possible effects on optical properties and screening of 2D objects, the above-mentioned reference is cited.

**The referee writes:**
_4) Slightly related to point 1 above: the ab initio 'validation' comes
only from the case of stanene. Considering the variation of results
of the models and the comparison with only one case, I would tone
down the validation. What is the amount of calculation here involved
(in terms of computational resources) and how much would be required to do an extra case ?
Here, I want to stress that I totally am in favour of the objective
of the article: to offer a much cheaper alternative (by using
models) to the ab initio brute force approach of exciton binding
energy. But a validation has to be meaningful._

**Our response:**
We understand the point arised by the Referee. An extra case would have been useful to further support the validation. However, silicene and germanene have spin-orbit gaps that are extremely small, a few meV. This creates severe problems of convergence and requires a very fine sampling of the Brillouin zone, thus requiring a huge amount of computational resources. Indeed, calculations for stanene  (with a DFT gap of about 77 meV) have been already very demanding, requiring around 40.000 CPU hours and the use of massive parallel supercomputers.
The idea is to investigate plumbene, although the honeycomb structure is not the most stable one. This fact, and the presence of an indirect gap, makes plumbene not an ideal test case, and for sure not a good case for possible comparison with experiments. Nevertheless, we plan to perform calculations on plumbene, but due to the very large computational cost, it will the subject of a separate future publication.

---

## Round 1 · Referee Report · Anonymous (Referee 2) · 2023-4-4

Report

The problem considered in the manuscript arXiv:2301.08601vI is interesting and recommend the manuscript "Transitions in Xenes between excitonic, topological and trivial insulator phases: influence of screening, band dispersion and external electric field" by Olivia Pulci et al., for publication.

---

## Round 2 · Referee Report · Anonymous (Referee 1) · 2023-5-4

Strengths

1) topic is important 2) approach is pertinent and (for the validation) state of the art

Weaknesses

1) presentation could have been more accurate

Report

In this revised submission, the authors meets the issues and points I raised in my report. Even though the paper has not really been restructured (as the authors claim) the corrections, extra sentences, more precise claims have improved the clarity of the article. As stated before, both the topic and the results obtained deserve publication. I can then confirm it here.

---

## Round 2 · Author Response

Hereby, we submit a revised version of the paper and we have already presented our detailed
response to the questions and criticisms raised by the referee. We believe that the present
version is suitable for publication in Scipost Physics

---

## Round 2 · List of Changes

1) In agreement with the suggestions of the referee we have improved the structure of the paper, the discussion of the results and the summary section in order to better clarify the physical content of the present analysis.

2) The physical meaning, advantages and limits of the various models discussed in the text are now better discussed in the manuscript.

3) The section with the ab initio results has been improved and, in particular, the approaches used to avoid the spurious interaction between stanene monolayers in neighboring supercell are now discussed in more details. Also, new references on this aspect have been introduced.

4) We have corrected some typos in the text.

---

## Editorial Decision

published